# R1 Vascular or Parenchymal Margins: What Is the Impact after Resection of Intrahepatic Cholangiocarcinoma?

**DOI:** 10.3390/cancers14205151

**Published:** 2022-10-20

**Authors:** Andrea Mabilia, Alessandro D. Mazzotta, Fabien Robin, Mohammed Ghallab, Eric Vibert, René Adam, Daniel Cherqui, Antonio Sa Cunha, Daniel Azoulay, Chady Salloum, Gabriella Pittau, Oriana Ciacio, Marc Antoine Allard, Karim Boudjema, Laurent Sulpice, Nicolas Golse

**Affiliations:** 1Department of Surgery, Paul-Brousse Hospital, Assistance Publique Hôpitaux de Paris, Centre Hepato-Biliaire, 94800 Villejuif, France; 2Department of Digestive, Oncological and Metabolic Surgery, Institute Mutualiste Montsouris, 75014 Paris, France; 3Department of Hepatobiliary and Digestive Surgery, CHU Rennes, 35000 Rennes, France; 4The Liver Unit, Queen Elizabeth Hospital Birmingham, Birmingham B15 2TH, UK; 5INSERM, Physiopathogénèse et Traitement des Maladies du Foie, UMR-S 1193, Université Paris-Saclay, 94805 Villejuif, France; 6INSERM, Equipe de Recherche « Chronothérapie, Cancers et Transplantation », Université Paris-Saclay, 94805 Villejuif, France

**Keywords:** intrahepatic cholangiocarcinoma, R1 resection, hepatectomy, prognosis, histopathology

## Abstract

**Simple Summary:**

The long-term outcome of R1 vascular (R1vasc) and R1 parenchymal (R1par) resections in the setting of intrahepatic cholangiocarcinoma (iCCA) is not well studied. Although the importance of the resection margin depth is clear, we aimed to clarify the impact of the R1 resection, by focusing on the outcomes between R0 resection and the two R1 types. The R1par resection presented a DFS and an OS intermediate between R0 and R1vasc. It appeared that a R1vasc resection should be avoided in patients with iCCA because it did not provide satisfactory oncological outcomes. Further studies could help to understand the best therapeutic procedure for these patients and the role of neo-adjuvant therapies in case of foreseeable R1vasc resection.

**Abstract:**

**Background**: to date, long-term outcomes of R1 vascular (R1vasc) and R1 parenchymal (R1par) resections in the setting of intrahepatic cholangiocarcinoma (iCCA) have been examined in only one study which did not find significant difference. **Patients and Methods**: we analyzed consecutive patients who underwent iCCA resection between 2000 and 2019 in two tertiary French medical centers. We report overall survival (OS) and disease-free-survival (DFS). Univariate and multivariate analyses were performed to determine associated factors. **Results**: 195 patients were analyzed. The number of R0, R1par and R1vasc patients was 128 (65.7%), 57 (29.2%) and 10 (5.1%), respectively. The 1- and 2-year OS rates in the R0, R1par and R1vasc groups were 83%, 87%, 57% and 69%, 75%, 45%, respectively (*p* = 0.30). The 1- and 2-year DFS rates in the R0, R1par and R1vasc groups were 58%, 50%, 30% and 43%, 28%, 10%, respectively (*p* = 0.019). Resection classification (HR 1.56; *p* = 0.003) was one of the independent predictors of DFS in multivariate analysis. **Conclusions**: the survival outcomes after R1par resection are intermediate to those after R0 or R1vasc resection. R1vasc resection should be avoided in patients with iCCA as it does not provide satisfactory oncological outcomes.

## 1. Introduction

Cholangiocarcinoma accounts for 3% of all digestive cancers [1,2,3]. Intrahepatic cholangiocarcinoma (iCCA) is the second most common primary liver tumor after hepatocellular carcinoma [4]. iCCA generally does not have a peritumoral capsule and therefore progresses via tissue infiltration, which usually extends to the main vascular-biliary trunks and the lymph node chains [4]. Tumor stage is often advanced at the time of diagnosis, which partly explains the poor prognosis.

Since chemotherapy has limited efficacy for iCCA [5,6], liver resection is the standard first-line treatment with curative intent [7]. Five-year survival rates after complete surgical resection range between 20% and 35% [8,9,10,11,12]. The standard strategy of iCCA resection is complete resection, in association with lymphadenectomy. A positive margin (≤1 mm, R1 resection) is associated with higher local recurrence rate and worse survival [11,12,13,14,15,16,17] than a negative margin (>1 mm, R0 resection); furthermore, disease-free survival (DFS) and overall survival (OS) incrementally improve as margin width increases [18]. R1 resection can be subdivided into 2 types, R1 parenchymal (R1par) and R1 vascular (R1vasc), which are defined in detail below [19,20,21,22,23]. Recent studies have confirmed the suitability of R1vasc resection in patients with colorectal liver metastases [24,25]. In the hepatocellular carcinoma setting, the gold standard remains complete removal of the tumor-bearing portal territory [26] with an adequate safety margin [27]; however, some authors have advocated that R1vasc resection is adequate [28].

Definition of the appropriate margins is a key question in oncological surgery, as tumor resectability and the sacrifice of surrounding parenchyma depend on it. Although the consequences of surgical margin width have been the subject of numerous publications over the past 30 years, the question of parenchymal versus vascular R1 resection in iCCA remains unclear. To the best of our knowledge, only one previous study has compared R1 resection types in iCCA patients. It concluded that R1vasc and R1par resection resulted in similar outcomes that were worse than R0 resection outcomes [29].

The present study aimed to elucidate the impact of R1 resection type (R1par and R1vasc) on oncological outcome in a large bicentric series of patients with iCCA, focusing on the distinction between types. Furthermore, we aimed to compare outcomes between R0 resection and the two R1 types.

## 2. Materials and Methods

We retrospectively examined patients who underwent liver resection for *mass-forming* iCCA between January 2000 and November 2019 in two French tertiary hepatobiliary centers. iCCA was histologically confirmed in all patients. We excluded patients in whom operative and/or pathological reports were not available for review. 

Surgical resection was retrospectively classified based on the operative and pathological reports as previously described [23,24,28]: R0 resection, was defined as margin width >1 mm; and R1 resection, margin width ≤1 mm. Three authors performed a comprehensive review of operative reports because the pathologists could not discriminate between R1vasc and R1par resection, as the vessels were not resected with the specimen. Any discrepancies were resolved by consensus. R1 resection was therefore subclassified as follows:R1vasc when the iCCA has been surgically detached from a hepatic vein (or a first or second order branch) or Glisson’s capsule (surrounding the branches of portal triad in its connective tissues), these structures having not been resected in a parenchymal-sparing strategy. R1vasc resection may potentially leave a microscopic tumoral residue in contact with the vascular structure.R1par when the parenchymal margin (distance between tumor and parenchymal section) was described as ≤1 mm.

Surgical resection was performed as previously described [19,20,21,22,23,24,25,28,30,31]. Intraoperative ultrasonography was routinely used. In patients with vascular contact, detachment of tumor from the vessels was considered only in cases of bilateral contact with a major vessel or vessels that precluded any possibility of complete resection or in cases where vascular resection was not possible for inflow/outflow reasons.

Adjuvant chemotherapy was administered according to national guidelines; in more recently operated patients, the BILCAP protocol was applied [32,33]. Patient follow up was performed every 3 months during the first two years, then twice a year. At each follow-up, tumor marker levels were evaluated, and computed tomography or magnetic resonance imaging was performed.

The OS was considered as interval between resection and last follow-up (death or alive) while the DFS was calculated as interval between resection and death, loss of follow-up or recurrence (first event). The postoperative morbidity was graded according to the Dindo-Clavien classification [34].

### Statistical Analysis

Quantitative data are expressed as means with standard deviation or medians with range. Qualitative data are expressed as numbers with percentage. The Kruskal–Wallis test was used to compare continuous variables between groups. Pearson’s chi-square or Fisher’s exact test was used to compare categorical variables as appropriate. Survival rates were calculated using the Kaplan–Meier method and compared using the log-rank test. Pathological and perioperative variables were assessed as predictors of outcomes (OS, DFS) using Cox proportional hazards models. Factors significantly associated with DFS were identified using univariable and multivariable Cox analyses. Predictors of R1vasc resection were assessed using logistical regression. Statistical analyses were performed using SPSS software version 20.0 (IBM Corp., Armonk, NY, USA). P. A *p*-value < 0.05 was considered significant.

## 3. Results

### 3.1. Preoperative Data 

A total of 195 consecutive patients who underwent curative liver resection for iCCA were included for analysis. Patient characteristics are summarized in Table 1. Preoperative characteristics did not significantly differ between the three groups except for male-to-female ratio (R0: 25.6%; R1par: 42.1%; and R1vasc: 50%; *p* = 0.04) and incidence of neoadjuvant external radiotherapy (R0: 2.4%; R1par: 0%; and R1vasc: 20%; *p* = 0.001). Median tumor size did not significantly differ between the groups (R0, 6 cm [range, 4.5–8.7]; R1par, 7 cm [range, 5–9.2]; and R1vasc, 7 cm [range, 3.5–10]; *p* = 0.16). The median values of total bilirubin, PT and platelets were of 8 [2–102] μmol/L, 91% [26–100] and 247 [76–100] * 103 G/L, respectively.

### 3.2. Peri-and Post-Operative Data

All the patients presented a *mass-forming* iCCA. As shown in Table 2 the number of R0, R1par and R1vasc patients were 128 (65.7%), 57 (29.2%) and 10 (5.1%) respectively. No patient was classified as mixed R1par and R1vasc and no patient underwent R2 resection. The iCCA was multifocal in 49 patients (25%) and lymph node-positive in 36 cases (18.5%). Major hepatectomy was required in 165 (85%). Twelve patients underwent laparoscopic resection. Among the 10 R1vasc group patients, tumors were detached from a portal branch in 6 cases and a hepatic vein in 4 cases. Compared to the R0 and R1par resection groups, operating time was significantly longer (*p* = 0.01) and proportion of patients who underwent biliary resection (R0, 15%; R1par, 21%; and R1vasc, 50%; *p* = 0.02) and portal vein reconstruction (R0, 8.5%; R1par, 3.5%; R1vasc, 30%; *p* = 0.02) was higher in the R1vasc group. Total number of lymph nodes resected during surgery was significantly higher in the R1vasc resection group (R0, 3 [range, 1–5]; R1par, 4 [range, 2–6]; R1vasc, 8 [range, 4–15]; *p* = 0.003). Total number of tumor nodules, tumor size, and proportion of tumors exhibiting microvascular and perineural invasion did not significantly differ between groups. Clinical and surgical features of the R1vasc patients are shown in Table 3.

### 3.3. Oncological Outcomes and Survival Results

Overall median follow-up was 29.7 months (range, 1–148). The 1- and 2-year OS rates in the R0, R1par and R1vasc groups were 83%, 87%, 57% and 69%, 75%, 45%, respectively (*p* = 0.296; Figure 1). The 3- and 5-year OS rates in the R0 and R1par groups were 56%, 49% and 39%, 22%, respectively; the corresponding rates in the R1vasc group were not calculated because of censoring.

The 1- and 3-year DFS rates in the R0, R1par and R1vasc groups were 58%, 50%, 30% and 35%, 25%, 10%, respectively (*p* = 0.019). Five-year DFS was not calculated because of censoring (Figure 2). Subgroup analysis of patients who received neoadjuvant chemotherapy (*n* = 30) showed similar results: DFS was shortest in the R1vasc group (*p* = 0.01). 

Location of recurrence significantly differed between the three groups (*p* = 0.013): the rate of intra-hepatic recurrence was higher in the R1par (76%) and R1vasc (83%) groups than R0 group (58%), whereas the rate of distant recurrence was higher in the R0 group. Subgroup analysis of patients who experienced recurrence showed that the repeat hepatectomy rate significantly differed between groups (R0 group, 24%; R1par group, 7.5%; and R1vasc group, 0%; *p* <0.05). Details are shown in Figure 3.

In univariate analysis, significant predictors of DFS were age (HR 1.02; *p* = 0.03), tumor size (HR 1.07; *p* = 0.002), resection type [R status] (HR 1.24; *p* = 0.01), number of lymph nodes invaded (HR 1.30; *p* = 0.003), number of lymph nodes resected (HR 1.05; *p* = 0.056), perineural invasion (HR 1.45; *p* = 0.04), microvascular invasion (HR 1.47; *p* = 0.02) and number of iCCA nodules (HR 1.33; *p* = 0.00002). Independent predictors of DFS in multivariate analysis were age (HR 1.02; *p* = 0.024), portal vein embolization (HR 2.4; *p* = 0.028), tumor size (HR 1.09; *p* = 0.037), resection classification (HR 1.56; *p* = 0.003), number of lymph nodes invaded (HR 1.31; *p* = 0.014) and number of iCCA nodules (HR 1.47; *p* = 0.001) (Table 4). In univariate analysis, significant predictors of OS were age (HR 1.01; *p* = 0.05), peritumoral secondary biliary cirrhosis (HR 1.57; *p* = 0.06); portal vein embolization (HR = 1.68; *p* = 0.06); tumor size (HR 1.05; *p* = 0.06), resection type [R status] (HR 1.03; *p* = 0.05), number of lymph nodes invaded (HR 1.30; *p* = 0.01), number of iCCA nodules (HR = 1.30; *p* = 0.002). Independent predictors of OS in multivariate analysis were portal vein embolization (HR = 4.09; *p* = 0.005) and number of lymph nodes invaded (HR 1.55; *p* = 0.001) (Table 5). 

In univariate analysis, 3 variables predicted the R1vasc status: neoadjuvant radiotherapy (OR = 16.3, 95%CI 2.4–111.8, *p* = 0.004), portal resection (OR = 6.2, 95%CI 1.4–26.8, *p* = 0.015) and biliary reconstruction (OR = 4.9, 95%CI 1.3–17.9, *p* = 0.016). In multivariate analysis, the neoadjuvant radiotherapy use remained statistically significant for R1vasc resection (OR = 13.6, 95%CI 1.7–108.1, *p* = 0.01).

## 4. Discussion

### 4.1. Statement of Principal Findings

Many publications have reported the impact of the R1vasc margin after resection of colorectal metastases, but only one study has investigated the matter in resection of iCCA. We present a homogeneous bicentric study that is representative of the current management of iCCA in western countries. To the best of our knowledge, this is the largest series to compare R0, R1vasc and R1par resection. The three studied groups had mostly similar characteristics before resection; however, a higher proportion of patients in the R1vasc group received preoperative external radiotherapy. Surgery in R1vasc patients was more demanding, as shown by longer operative time and higher rates of portal and biliary reconstruction, which resulted in a higher rate of significant postoperative morbidity.

We demonstrated that oncological outcome in patients with iCCA is independently associated with resection classification. R0 resection was significantly superior to R1par and R1vasc resection in terms of DFS. Moreover, 1- and 2-year OS tended to be poorer in the R1vasc group, although not significant because of small sample size; the type of resection status being a significant predictor for OS in univariate analysis. The lesser prognostic impact of R1vasc resection on OS could be partly explained by the fact that these patients, after recurrence, had access to chemotherapy in 67% of cases, compared with only 52% for R0/R1par patients. This may level out the long-term curves. The high rate of intrahepatic recurrence after R1 resection did not allow increased access to curative treatment because these recurrences were frequently associated with distant recurrence.

Our analysis showed that predictors for R1vasc resections, on preoperative imaging, were the presence of vascular or biliary proximity of the tumor requiring resection (a rather infrequent occurrence in iCCA), and the absence of vascular/glissonian intraoperative detachment despite neoadjuvant radiotherapy. In fact, a tumor originating in contact with the hepatic vessels may be more aggressive on account of its location alone, making it easier for neoplastic cells to spread. 

### 4.2. Interpretation with Reference to Other Studies

Previous studies have reported that resection margin ≤1 mm is associated with higher local recurrence rate and worse outcome [11,12,13,14,15,16,17,18]. Our study aimed to analyze survival and recurrence according to resection classification. The proportion of R1 patients in our study was 34%, including R1par (29%) and R1vasc (5%), which is similar to proportions reported in previous publications [29,35]. In these studies, R1 resection rates ranged between 29% and 32%.

In a recent iCCA study of 59 patients, including 17 (29%) who underwent R1vasc resection, Torzilli et al. reported that risk of local recurrence was similar in the R1vasc (29%) and R1par groups (36%), but lower in the R0 group (3%, *p* = 0.003) [29]. The R1vasc and R1par groups had similar median OS (30 months vs. 30 months) and median DFS (10 months vs. 8 months), which were significantly lower than the corresponding rates in the R0 group (70 and 39 months, respectively; *p* = 0.066 and *p* = 0.007, respectively). They suggested that R1vasc resection could be considered to achieve resectability in otherwise unresectable patients because the vasculature acts as a barrier to tumor diffusion. However, the study’s small sample size and high proportion of patients who underwent R1vasc resection does raise questions about the external validity of their results.

Our results contradict the suggestion of Torzilli et al. because oncological outcomes were worse in our R1vasc patients than in the R1par and R0 patients. Moreover, our findings are more in line with the disease’s pathophysiology. E-cadherin is a transmembrane glycoprotein involved in cell-cell adhesion between tumor epithelial cells and vascular endothelial cells and thus relevant to tumor spread to vessels. Colorectal cancer metastases do not have the ability to spread into vessels because of high E-cadherin protein expression. In contrast, iCCA has a low level of E-cadherin expression so cellular adhesion is attenuated; thus, it can readily enter vessels [36,37]. This may explain, at least partly, why outcomes were worse in iCCA tumors contacting the vasculature. Further studies are warranted to assess this hypothesis. 

The poor results observed in our R1vasc group are in accordance with the work of Jia et al., who examined the role of post-operative radiotherapy in iCCA patients who underwent R1vasc resection [38]. In their surgery alone groups, R1vasc patients presented median OS and DFS was 15 months and 5.5 months, respectively, which reinforces the validity of our findings [38].

Despite the emergence of novel therapeutic options, surgery remains the best curative treatment. However, results of R1vasc resection are not encouraging, as only one patient survived longer than 24 months. We therefore do not recommend it, unless integrated in a multidisciplinary approach. Conversely, surgery offered better survival than medical management for R1par and R0 patients. 

Patients with advanced iCCA (strictly intrahepatic iCCA with high tumoral load and/or near the vasculature) may be downstaged and then subsequently considered suitable for surgery. In these patients, every effort should be made to obtain satisfactory vascular margins when possible and extended hepatectomy may be required. Preoperative portal vein embolization may be necessary to allow for an increased extent of resection and enable resection of invaded pedicles or veins. We recently reported the use of neoadjuvant chemotherapy for downstaging unresectable iCCAs and observed OS and DFS identical to those observed in patients with initially resectable tumors [39]. The use of neoadjuvant therapy, particularly radioembolization, for downstaging when an R1 resection is foreseeable and unavoidable is, in our opinion, a strategy that should be pursued with more determination to achieve better surgical results [40]. In our experience, neo-adjuvant radiotherapy was not, per se, a risk factor for worse results (3/5 patients resected R0); however, in the absence of vascular detachment, it was associated with R1vasc status and worse DFS. Interesting also seems to be the use of intra-arterial yttrium-90 radioembolization combined with systemic chemotherapy to downstage tumor lesions to avoid R1 resections [41]. A prospective randomized French study is currently assessing the impact of neoadjuvant selective internal radiation therapy + capecitabine in resectable iCCA (ongoing Sirocho study, NCT05265208).

In cases of a foreseeable R1vasc resection (foreseeable vasculo-biliary resection, failure of vascular detachment following radiotherapy), non-surgical management should be considered as the first-line treatment, as the 1- and 2-year OS rates in the R1vasc group were 57% and 45%, respectively, which are not better than the rates reported for patients undergoing radioembolization plus chemotherapy alone [40]. Positron emission tomography/computed tomography (PET/CT) may also play a role in lymph node involvement for locally advanced iCCA [42]. While not assessed in this report, the role of PET/CT for nodal staging could also be of great interest [43]. Indeed, the presence of suspicious lymph nodes on PET could be a negative predictor, particularly if associated with R1vasc resection [44].

If R0 or R1par resection would not be possible (with or without downstaging), palliative treatment should be strongly considered. R1vasc resections not only provide unsatisfactory long-term overall and recurrence-free survival, but the presence of severe postoperative morbidity is more frequent. This is probably because of the association of biliary and/or vascular reconstructions with R1vasc resections. In addition, patients with a recurrence after R1vasc resection cannot benefit from a repeat hepatectomy, thus limiting the oncologic benefit of such a “palliative” procedure. One possible avenue to study further would be the administration of adjuvant radiotherapy, which has been specifically tested in R1vasc patients (2 uncontrolled pilot studies): [38,39,40,42,43,44,45]. 

We have already published a study concerning preoperative predicting postoperative risks after resection of perihilar cholangiocarcinoma; in this series, we had 61 cases of iCCA, and the predictors of severe morbidity were: male gender, portal vein embolization, planned biliary resection, low psoas muscle area/height^2^ and low hemoglobinemia [46]. As a future prospective, we wish to establish a nomogram with the aid of Artificial intelligence to predicts not only post-operative complications, but also the R1 resection rate.

### 4.3. Study Limitations

The patient follow-up was relatively short, and the sample size was small, which prevented us from drawing any definitive conclusions. Propensity score analysis could not be performed because of the small sample size. Nevertheless, our study reflects modern western management of iCCA in two high-volume tertiary centers, with homogeneous population in terms of preoperative characteristics. The limited number of patients in the R1vasc group did not allow a subgroup analysis of the differences between R1-hepatic vein or R1-glissonean pedicle, both included in R1vasc group. Certainly, these contacts may bring different consequences for technical resectability and oncological outcomes and should be explore in further studies. Li et al. reported a different oncological impact of peri-portal vs. peri-hepatic vein location of iCCA (*n* = 352), but the absence of reported R1 patients prevents direct comparison with our results [46].

Furthermore, lymphadenectomy was not performed systematically in approximately 46% of patients, which is consistent with other data in the literature [47], thus preventing to analyze the impact of lymph node involvement. 

We were unable to analyze preoperative imaging features, particularly precise intra-hepatic iCCA location, because this data was not systematically recorded. In addition, because modifications of neoadjuvant chemotherapy regimens occurred over the 20-year period in which the study was conducted, we were unable to accurately assess pathological response and the effect of chemotherapy on resection status [48].

## 5. Conclusions

After R0, R1par or R1vasc resections of iCCA, the survival was worst in R1vasc resection group, better in R0 group. Although the OS curve was non statistically different over 5 years, it remains impressive that the 3-year survival of the R1vasc group is 0%. The small number of R1vasc patients does not allow definitive conclusions to be drawn but, in view of ours the results, it does not invalidate the relevance of the question. If R1vasc resection is predicted, based on preoperative imaging (e.g., a central lesion requiring vasculo-biliary resection), every effort should be made to reduce the tumor size or vascular contact using neoadjuvant therapy (Y90 +/− chemotherapy, external radiotherapy). Resection should probably be avoided in cases in which predicted margins cannot be improved (no vascular detachment). In these cases, the interest in offering surgical resection should be probably questioned because of the poor oncological outcomes.

## Figures and Tables

**Figure 1 cancers-14-05151-f001:**
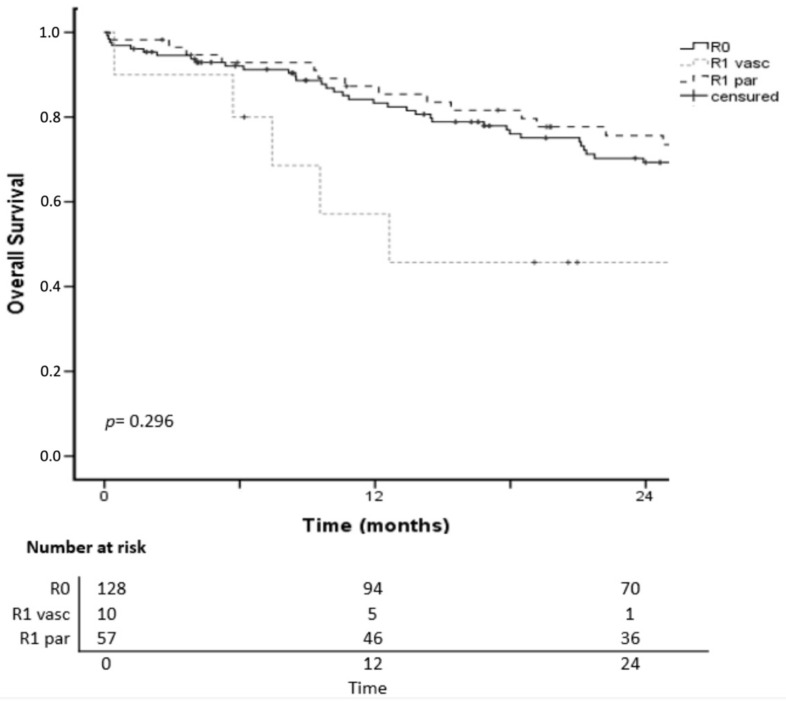
Overall survival after hepatectomy for iCCA according to R status.

**Figure 2 cancers-14-05151-f002:**
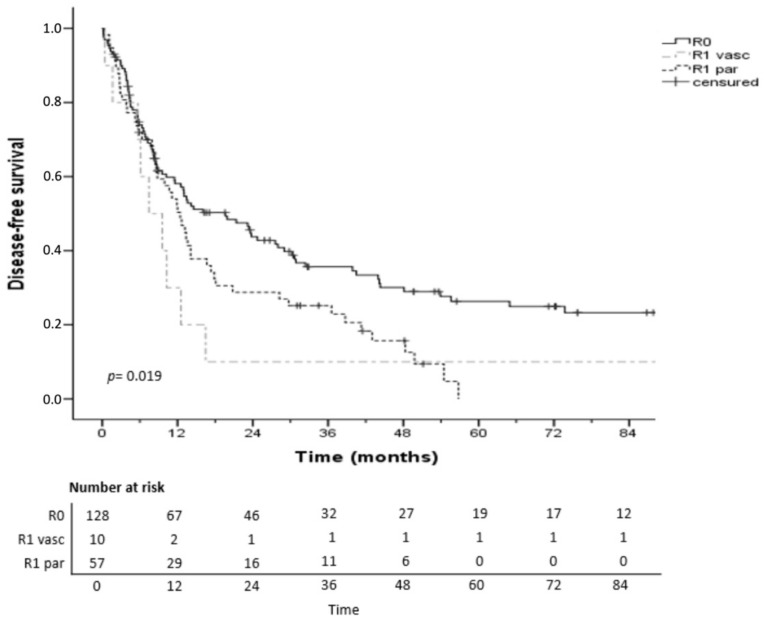
Disease-free survival after hepatectomy for iCCA according to R status.

**Figure 3 cancers-14-05151-f003:**
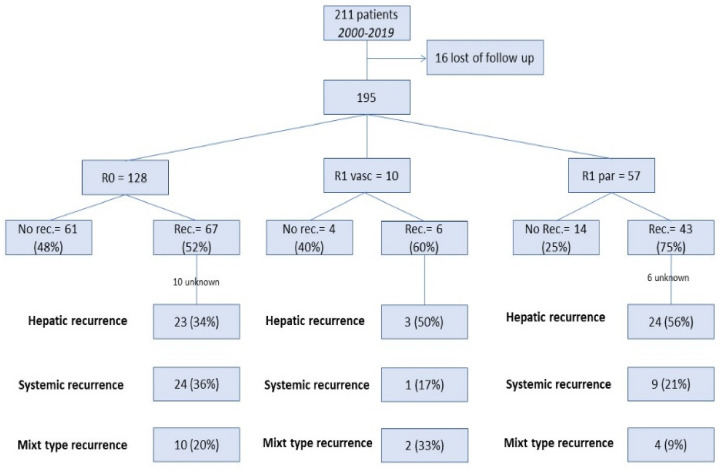
Outline of study results.

**Table 1 cancers-14-05151-t001:** Preoperative features according to R status.

	R0 (*n* = 128)	R1 Vasc. (*n* = 10)	R1 Par. (*n* = 57)	*p* Value
Age (years)	67 [60–72]	71 [69.2–75.0]	64 [58–73]	0.16
BMI (Kg/m^2^)	25.5 [22.5–28.5]	25.2 [21.3–27.2]	25.8 [22.8–28.5]	0.89
Tumor size (cm)	6 [4.5–8.7]	7 [3.5–10]	7 [5–9.2]	0.16
Male gender	33 (25.6)	5 (50)	24 (42.1)	0.04
Peritumoral secondary biliary cirrhosis	26 (20.2)	2 (20)	4 (7)	0.07
Preop. biliary drainage	7 (5.5)	2 (20)	3 (5.4)	0.18
Portal vein embolization	16 (12.4)	1 (7)	4 (10.7)	0.54
Neoadjuvant radiotherapy	3 (2.4)	2 (20)	0 (0)	0.001
Neoadjuvant chemotherapy	22 (17.2)	2 (20)	6 (10.5)	0.50

*n* = number [ ] = range; ( ) = percentage.

**Table 2 cancers-14-05151-t002:** Peri- and postoperative data according to R status.

	R0 (*n* = 128)	R1vasc. (*n* = 10)	R1par. (*n* = 57)	*p* Value	R0 vs. R1vasc	R1vasc vs. R1par
Operative time (min)	215 [158–316]	345 [150–554.2]	299 [180–390.5]	0.01	0.12	0.47
Liver pedicle clamping	28 [15–45]	30 [0–59]	40 [7–60]	0.12	0.77	0.84
Major liver resection	102 (81)	9 (90)	46 (82.1)	0.77	0.50	0.53
Biliary resection	19 (14.7)	5 (50)	12 (21.1)	0.02	0.04	0.05
Arterial resection	3 (2.3)	0 (0)	2 (3.5)	0.78	0.62	0.54
Portal vein resection	11 (8.5)	3 (30)	2 (3.5)	0.02	0.03	0.003
Lymph nodes analyzed	3 [1–5]	8 [4–15.2]	4 [2–6]	0.003	0.01	0.057
Patients N+	21 (16.3)	4 (40)	11 (19.3)	0.25	0.06	0.012
Patients Nx	58 (45)	4 (40)	20 (35.1)	0.25	0.14	0.22
Numbers of tumors	1 [1,1]	1 [1–4.2]	1 [1,1]	0.16	0.23	0.40
Max tumor size (cm)	6 [4.50–8.75]	7 [3.50–10.00]	7 [5.00–9.25]	0.16	0.73	0.73
Perineural invasion	27 (21.1)	5 (50)	15 (26.3)	0.10	0.03	0.13
Microvascular invasion	53 (41.4)	6 (66.7)	30 (52.6)	0.16	0.13	0.43
Clavien Dindo ≥ 3	29 (25)	5 (50)	12 (21)	0.13	0.06	0.06

[ ] = range; ( ) = percentage.

**Table 3 cancers-14-05151-t003:** Clinical and surgical features of the R1vasc patients.

R1vasc (Age, Sex)	R1 location (Vascular Contact)	Type Hep	Biliary Resec	Portal Resec	Relapse	Death at Follow-Up	OS (Mounths)	DFS (Mounths)	N (+/tot)
A 71 y, F	Hepatic vein	Left Hep	Yes	No	Yes	Yes	12.6	1.6	1/16
B 71y, F	Hepatic vein	Right Hep	Yes	Yes	Yes	No	6.2	6.1	2/6
C 71 y, M	Portal Vein	Left Hep	Yes	No	Yes	No	19.0	16.5	4/15
D 72 y, M	Portal Vein	Right Hep	Yes	Yes	No	Yes	7.4	7.4	0/10
E 64 y, F	Portal Vein	Left Hep	No	No	Yes	No	20.9	10.2	2/5
F 58 y, F	Hepatic vein	Minor Hep	No	No	Yes	No	20.6	12.5	-
G 75 y, M	Portal Vein	Right Hep	No	No	No	Yes	9.6	9.6	-
H 75 y, M	Hepatic vein	Left Hep	No	No	No	Yes	0.4	0.4	0/1
I 74 y, M	Portal Vein	Right Hep	Yes	Yes	No	Yes	5.7	5.7	-
L 76 y, F	Portal Vein	Right Hep	No	No	Yes	Yes	127.1	113.9	-

y = years, F = female, M = male, DFS = disease free survival, OS = overall survival, N = lymph node number (positive/total).

**Table 4 cancers-14-05151-t004:** Univariate and multivariate analysis for DFS.

Variable		Univariate Analysis			Multivariate Analysis	
	HR	CI	*p* Val.	HR	CI	*p* Val.
Gender (female)	1.07	0.76–1.50	0.69			
Age	1.02	1.00–1.03	0.03	1.02	1.00–1.05	0.024
BMI	0.98	0.95–1.02	0.39			
Peritumoral secondary biliary cirrhosis	1.10	0.74–1.65	0.62			
Major liver resection	1.22	0.79–1.86	0.35			
Preop. biliary drainage	1.54	0.85–2.79	0.17			
Portal vein embolization	1.47	0.93–2.31	0.09	2.4	1.10–5.28	0.028
Neoadjuvant chemotherapy	1.32	0.87–2.02	0.19			
Neoadjuvant radiotherapy	1.00	0.36–2.70	1.00			
Tumor size	1.07	1.03–1.13	0.002	1.09	1.00–1.19	0.037
R0, R1vasc, R1par	1.24	1.05–1.47	0.01	1.56	1.16–2.09	0.003
Numbers of lymph nodes invaded	1.30	1.09–1.55	0.003	1.31	1.05–1.65	0.014
Numbers of lymph nodes resected	1.05	0.99–1.10	0.056	1.004	0.93–1.09	0.92
Perineural invasion	1.45	1.02–2.07	0.04	1.16	0.59–2.27	0.66
Microvascular invasion	1.47	1.06–2.03	0.02	0.75	0.39–1.40	0.35
Number of iCCA nodules	1.33	1.16–1.51	<0.001	1.47	1.18–1.83	0.001
Biliary resection	0.94	0.62–1.43	0.76			
Arterial resection	0.78	0.25–2.46	0.6			
Portal vein resection	0.96	0.53–1.73	0.96			

**Table 5 cancers-14-05151-t005:** Univariate and multivariate analysis for OS.

Variable		Univariate Analysis			Multivariate Analysis	
	HR	CI	*p* Val.	HR	CI	*p* Val.
Gender (female)	1.22	0.81–1.83	0.32			
Age	1.01	1.00–1.03	0.05	1.00	0.98–1.03	0.64
BMI	1.00	0.96–1.07	0.86			
Peritumoral secondary biliary cirrhosis	1.57	0.98–2.5	0.06	1.92	0.67–5.48	0.21
Major liver resection	0.89	0.59–1.57	0.89			
Preop. biliary drainage	1.04	0.44–2.37	0.92			
Portal vein embolization	1.68	0.97–2.91	0.06	4.09	1.52–10.98	0.005
Neoadjuvant chemotherapy	1.28	0.76–2.15	0.36			
Neoadjuvant radiotherapy	0.86	0.21–3.52	0.83			
Tumor size	1.05	1.00–1.11	0.06	1.03	0.92–1.15	0.59
R0, R1vasc, R1par	1.03	1.00–1.12	0.05	1.16	0.79–1.71	0.44
Numbers of lymph nodes invaded	1.30	1.05–1.61	0.01	1.55	1.19–2.02	0.001
Numbers of lymph nodes resected	1.00	0.94–1.07	0.80			
Perineural invasion	1.18	0.77–1.82	0.44			
Microvascular invasion	1.03	0.70–1.52	0.85			
Number of iCCA nodules	1.30	1.10–1.15	0.002	1.33	0.93–1.91	0.11
Biliary resection	0.65	0.37–1.14	0.19	0.85	0.37–1.95	0.71
Arterial resection	0.77	0.26–2.66	0.76			
Portal vein resection	1.20	0.60–2.38	0.60			

## Data Availability

The data presented in this study are available on justified request.

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
