# Peer review of "R1 Vascular or Parenchymal Margins: What Is the Impact after Resection of Intrahepatic Cholangiocarcinoma?"

_cancers, 2022, doi:10.3390/cancers14205151_

Round 1

Reviewer 1 Report

1. p8, L20 and L24, need to code the references  about the post-operative complication. 

2. Please make more short about the section of "Limitation"

Author Response

Reviewer 1

Point 1. p8, L20 and L24, need to code the references about the post-operative complication.

Dear reviewer, we are sincerely sorry, but we do not understand where you want us to add a reference about postoperative complications. Indeed, page 8 is the discussion and lines 20-24 do not mention complications. For more clarity, we have added a sentence explaining the grading according to Dindo-Clavien in the “Materials and methods” section and we added the corresponding reference. If this does not meet your request, we can review this change.

Point 2. Please make more short about the section of "Limitation"

We have shortened this chapter slightly, while still leaving the limitations of this work clearly displayed, in the interest of intellectual honesty.

Reviewer 2 Report

The authors showed unsatisfactory oncological outcomes of R1 resection focusing on outcomes between R0 resection and the two R1 types. They concluded that R1vascular resection should be avoided in patients with intrahepatic cholangiocarcinoma (iCCA).

I think it is not suitable for the Cancers because of the small number of patients who underwent R1 vascular, poor preoperative analysis and data collection, and short follow-up period.

Major:

1. Preoperative analysis was not enough. You know, iCCA is classified into mass-forming, periductal infiltrating, or combined types. The classification is closely associated with the proportion of lymph node metastasis and distant metastasis, and even the outcomes after resection. The authors must show the location of the tumors, and the classification.

2. R1 vascular means the tumor was surgically detached from a hepatic vein or Glissonean capsule. The author should show which vasculature the tumors attached to.

3. What is the "position" in the Table 3.  The tumors were detached from the vena cava? It's different from a hepatic vein...

4. The preoperative data of the R1 vascular is much worse than R0 and R1 parenchymal. As the author mentioned, PSM is required with a large number of patients.

5. R1 vascular is significantly associated with better RFS, but not OS. Why do you think this result comes?

6. The authors should elucidate the location of recurrence, especially intrahepatic recurrence. Intrahepatic metastasis or local recurrence is completely different. The local recurrence can occur in the R1 vascular resection. When a patient has only a local recurrence, it may be resected. However, Intrahepatic multiple recurrence or distant metastasis cannot be resected.   

Minor

1. The title of Figure 2 has a mistake. OS → DFS

Author Response

Reviewer 2

The authors showed unsatisfactory oncological outcomes of R1 resection focusing on outcomes between R0 resection and the two R1 types. They concluded that R1vascular resection should be avoided in patients with intrahepatic cholangiocarcinoma (iCCA).

I think it is not suitable for the Cancers because of the small number of patients who underwent R1 vascular, poor preoperative analysis and data collection, and short follow-up period.

Dear reviewer, thank you for your careful review. We are fully aware of the limitations of this work, and we have clearly detailed them in the discussion. However, when reading the data already published, we believe that this work is original and can lead to a real impact on the daily management of these patients. Indeed, at a time when neo-adjuvant treatments are becoming more and more frequent, the identification of subgroups at risk seemed to us useful.

Major:

Point 1. Preoperative analysis was not enough. You know, iCCA is classified into mass-forming, periductal infiltrating, or combined types. The classification is closely associated with the proportion of lymph node metastasis and distant metastasis, and even the outcomes after resection. The authors must show the location of the tumors, and the classification.

Thank you for this pertinent comment.

Regarding the histological subtype, all tumors were mass-forming, with a median tumor size over 6 cm. The periductal-infiltrating and intraductal-growth types were not included in this work as they can usually be suspected on preoperative imaging, respectively with a growth along the bile duct without mass formation, and diffuse and marked ductectasia with or without a grossly visible papillary mass or a focal stricture-like lesion with proximal ductal dilatation. Because the mass-forming type of iCCA is the most common morphological subtype (>85% of cases), this selection is relevant for the reader. We have therefore clarified this important aspect in chapters 2 and 3.2.

Concerning the tumor location (within the liver parenchyma of iCCA), there is no consensus classification. Indeed, we analyzed this data, by looking at whether the tumors had a peripheral, central, close to the portal bifurcation, close to the cavo-hepatic confluence or close to the inferior vena cava location. Unfortunately, this distinction did not predict the type of R resection, and thus risked adding unnecessary confusion to the overall message. We therefore preferred not to specify this point.

Point 2. R1 vascular means the tumor was surgically detached from a hepatic vein or Glissonean capsule. The author should show which vasculature the tumors attached to.

We do not understand this remark well since it seems to us that this distinction has been already specified in table 3 and paragraph 3.2. Among the 10 R1vasc group patients, tumours were detached from a portal branch in 6 cases and a hepatic vein in 4 cases. We remain at your disposal in case of misunderstanding on our part.

Point 3. What is the "position" in the Table 3.  The tumors were detached from the vena cava? It's different from a hepatic vein...

Indeed, this term was not very clear. We have changed it to "R1 location (vascular contact)" and corrected the data entered. 

Point 4. The preoperative data of the R1 vascular is much worse than R0 and R1 parenchymal. As the author mentioned, PSM is required with a large number of patients.

Dear reviewer, we find your judgment a bit excessive in that the main preoperative difference between the three groups was the percentage of neoadjuvant radiotherapy. Gender difference had no real clinical impact. It did not seem possible to us to make a propensity score, for at least two reasons:

-first, from a theoretical point of view, the already limited number of patients would have resulted in a too small final population;

-secondly, and more importantly, the PSM would not have allowed the selection of more comparable groups by taking the R1vasc group as a baseline. Indeed, no model would have allowed the selection of R0 and R1vasc patients who had received neoadjuvant radiotherapy, since these patients are indeed very rare. By taking the R0 or R1par group as reference, the PSM would have led to a drastic reduction in the R1vasc population, making any interpretation impossible. Therefore, even if it was a statistical solution initially considered, we do not consider useful to propose a PSM in this particular work.

Regarding the enlargement of the population, we included the two French centers with the most experience in the field. The inclusion of other centers would have led to an increase in missing data, given the large inclusion period considered.

Point 5. R1 vascular is significantly associated with better RFS, but not OS. Why do you think this result comes?

Dear reviewer, we believe there is a misinterpretation of our results. Indeed, the R1vasc group clearly showed a lower DFS than the other two groups. Regarding OS, it is true that the curves were not statistically different, but the curve of the R1vasc group was clearly lower. A larger cohort would certainly have shown a statistical difference.

In our opinion, one explanation could be the accessibility of systemic treatment in case of recurrence. Thus, R1vasc patients with recurrence had access to chemotherapy in 67% of cases, compared with only 52% for R0 and R1par patients. This may level out the long-term curves. More generally, survival after oncologic surgery is always multifactorial and is therefore a less relevant endpoint than DFS. We have added this hypothesis to the discussion.

Point 6. The authors should elucidate the location of recurrence, especially intrahepatic recurrence. Intrahepatic metastasis or local recurrence is completely different. The local recurrence can occur in the R1 vascular resection. When a patient has only a local recurrence, it may be resected. However, Intrahepatic multiple recurrence or distant metastasis cannot be resected.   

We fully agree with you but it seems to us that this has been addressed in our work.  As already illustrated in Fig 3, our results show that a R1 resection has a much higher risk of intrahepatic recurrence than an R0 resection, which is in line with the results found in the literature.

Intrahepatic recurrence occurred in 58% of recurrence after R0 resection, 83% in R1vasc and 76% of R1par.

However, this intrahepatic recurrence was not always exclusively intrahepatic, which ultimately did not allow effective local treatments to be considered more often.

We have added numerical data in the results section and a brief comment in the discussion.

Minor

Point 1. The title of Figure 2 has a mistake. OS  DFS

Yes, thanks for this careful review. We modified the title.

Reviewer 3 Report

I read with interest this manuscript, where the authors compare the results of R0, R1vasc, and R1par liver resections for iCCA. The Rvasc group had a DFS rate of 28% at 1 year, and the authors discourage this approach. Overall the manuscript is well structured and well written. The results quite well support the conclusions.

The main limitation is the lack of systematic lymphadenectomy, which is however well acknowledged in the discussion.

I have only a few minor comments 

1) Table 1, “cirrhosis”. This seems too generic, please specify the Child-Pugh class (with points).

2) The resolution of figures 1 and 2 should be far improved.

3) Figure 2 (caption): this is DFS and not OS.

4) p. 10, lines 306-7 (Propensity score…): this clarification is not necessary and this sentence can be removed.

Author Response

Reviewer 3

I read with interest this manuscript, where the authors compare the results of R0, R1vasc, and R1par liver resections for iCCA. The Rvasc group had a DFS rate of 28% at 1 year, and the authors discourage this approach. Overall, the manuscript is well structured and well written. The results quite well support the conclusions.

The main limitation is the lack of systematic lymphadenectomy, which is however well acknowledged in the discussion.

Thank you very much for your encouraging remarks. Concerning lymphadenectomy, we were also disappointed by this finding, but this is (unfortunately) in complete agreement with the recent French review published by Hobeika et al. in BJS 2021.

I have only a few minor comments 

Point 1. Table 1, “cirrhosis”. This seems too generic, please specify the Child-Pugh class (with points).

Thank you for your very pertinent comment. None of our patients had a cirrhosis. It was simply a peritumoral biliary compression responsible for secondary biliary cirrhosis. We have changed the wording throughout the text.

Point 2. The resolution of figures 1 and 2 should be far improved.

We modified these figures.

Point 3. Figure 2 (caption): this is DFS and not OS.

It’s DFS, we have made the correction in the text. Thanks for careful reading.

Point 4. p. 10, lines 306-7 (Propensity score…): this clarification is not necessary and this sentence can be removed.

Dear reviewer, we think this discussion is necessary because it is a comment from reviewer 2 and the reader might also wonder. We therefore prefer to maintain this rationale. Thank you for your understandin
